# Influences of Slenderness and Eccentricity on the Mechanical Properties of Concrete-Filled GFRP Tube Columns

**DOI:** 10.3390/polym13172968

**Published:** 2021-08-31

**Authors:** Hongbo Guan, Yifei Xia, Jinli Wang, Arsene Hugo Mbonyintege

**Affiliations:** School of Civil Engineering, Liaoning Technical University, Fuxin 123000, China; yifeixia97@gmail.com (Y.X.); wangjinli@lntu.edu.cn (J.W.); xiagurenxin001@163.com (A.H.M.)

**Keywords:** GFRP tube, confined concrete, eccentricity, slenderness ratio

## Abstract

The existence of either eccentricity or slenderness has a significant effect on the mechanical properties of a structure or member. These properties can change the working mechanism, failure mode, and bearing capacity of the structure or member. A concrete-filled, glass fibre-reinforced, polymer tube composite column has the same problem. We carried out experiments on the influences of eccentricity and slenderness on the mechanical properties of concrete-filled, glass fibre-reinforced, polymer tube composite columns. The experimentally recorded stress–strain relationships are presented graphically, and the ultimate axial stresses and strains and the FRP tube hoop strains at rupture were tabulated. The results indicate that the influences of slenderness and eccentricity on the composite columns were significant with regard to the axial strain, hoop strain, ultimate bearing capacity, lateral displacement, and failure mode. Based on the existing research literature and the results reported in this paper, the bearing capacity formula of a composite slender column under an eccentric load was established. The theoretical results were in good agreement with the experimental results.

## 1. Introduction

Over the past couple of decades, fibre-reinforced polymer (FRP) composites have found increasingly more applications in civil engineering due to their high strength-to-weight ratios and high corrosion resistance [1,2]. Existing reinforced concrete structures are repaired with FRP materials [3,4,5,6]. With an external FRP wrap, an RC column’s ductility, axial load, moment capacity, and energy absorption are enhanced because of the external passive confinement which depends on the lateral expansion of the concrete as a response to axial loading [7]. Recently, the concrete-filled FRP tube (CFFT) was introduced for efficient use of FRP reinforcement for new columns. The CFFT technique for new column construction was investigated in the literature as a practicable alternative of the steel RC column. Concrete-filled FRP tubes (CFFTs) provide a new and attractive way to use composite materials for several applications, including piles, columns, bridge piers, poles, and overhead highway sign supports. FRP plays an important role in these structures by providing lateral confining pressure that controls the volumetric expansion of the concrete to increase their ultimate load capacities and enhance their structural ductility.

A number of experimental studies have been conducted to investigate the axial compressive behaviour of CFFTs over the last two decades. Under axial loads, the FRP tube confines the concrete by reducing its lateral expansion, and therefore it increases its ultimate strain and strength [8,9,10].

In addition, concrete compression columns almost do not exist in practical engineering. Several researchers have studied the structural behaviour of concrete-filled FRP tubes under eccentric loads. The lateral deflection caused by longitudinal bending under an eccentric load causes premature buckling and failure of composite columns. Therefore, the effects of the secondary moment caused by lateral deflection on the mechanical properties of composite columns cannot be ignored [11]. The flexural behaviour of members was also studied by Hadi et al. [12] (glass FRP tubes) and by Gholampour et al. [13] (carbon-FRP tubes). Fam et al. investigated the properties of concrete-filled cross-sectional configurations, including a tube with a central hole, tube-in-tube with concrete filling in between, and GFRP tubes with different laminate structures. The results indicated that the flexural behaviour is highly dependent on the stiffness and diameter-to-thickness ratio of the tube, and to a much lesser extent, on the concrete strength [14]. Mirmiran et al. studied both thin and thick-walled tubes to investigate under- and over-reinforced sections subjected to a constant axial load and increasing bending, using transverse loads, in order to compare the behaviour of concrete-filled FRP tubes with that of conventional prestressed piles. The study concluded that bond failure was not a concern in members subjected to combined bending and axial loads, and over-reinforced sections were recommended due to their greater strength and stiffness [15]. G. Lin and J. G. Teng presented new details of the behaviour of FRP-confined concrete in an eccentrically loaded circular column based on results obtained from a 3D FE investigation. The proposed model can be directly used in section analysis or in a theoretical column model, and has been shown to provide much more accurate predictions for the ultimate displacement/curvature of test columns than any existing CL stress strain model [16].

It is well known that most columns in practice are subjected to combined axial compression and bending due to load eccentricities. Additionally, the effect of slenderness on the compressive behaviour of a column is an important issue to be addressed. Many experimental studies have been conducted on slender FRP-confined concrete columns. These studies have confirmed that FRP confinement can enhance the strength and ductility of slender columns. It has also been reported in these studies that an increase of column slenderness generally leads to a reduction in the load capacity of the column [17].

Maha Hussein Abdallah et al. presented the test results of an experimental program to investigate the structural performance of slender, concrete-filled, fibre-reinforced polymer (FRP) tube (CFFTs) columns under pure axial compression loads. The experimental results showed that the axial compressive strength of CFRP-reinforced CFFT columns was reduced by 22% when the slenderness ratio was increased from 8 to 20. In general, the use of an FRP tube induces a confinement effect on the concrete column which enhances its strength and ductility. However, the confinement results in slenderness, which increases the possibility of buckling instability in CFFT columns [18]. Jason Fitzwilliam et al. studied the effects of slenderness on carbon FRP-wrapped circular RC columns under eccentric axial loads. It was shown that CFRP wraps increase the strength and deformation capacities of slender columns, but the beneficial confining effects were proportionally greater for short columns [19]. Masood NoroozOlyaee and Davood Mostofinejad investigated the behaviour of slender RC columns retrofitted with fibre-reinforced polymer (FRP) composites. Nine circular concrete columns with slenderness ratios ranging from 15.4 to 27.7 were tested. The results showed that with increasing slenderness, load-carrying capacity decreased [20]. In Yu T. et al., a new type of composite structure was introduced. The test results confirmed excellent performance of the slender, fibre-reinforced polymer (FRP)-confined, concrete-encased, cross-shaped steel columns (FCCSCs), and showed that the load capacity of FCCSCs decreases with the slenderness ratio and the load eccentricity [21].

As mentioned previously, a large number of experimental studies on FRP-reinforced columns have concentrated on the influences of slenderness and eccentricity. Even if the parameters were different in each study, the facts indicate that FRP can significantly improve the mechanical properties of columns [22]. Increases in column slenderness and eccentricity generally lead to reductions in the load capacity of the column. 

To date, research related to quantitative evaluations of the influences of slenderness ratio and eccentricity on the bearing capacity of FRP-confined concrete columns is still very limited, especially for GFGT composite columns, which are considered the most likely to be used in engineering practice. One reason is there are many unsolved problems in FRP-confined concrete short columns; the other is that the effect of column slenderness (i.e., the second-order effect) is not included in various design guidelines for the FRP strengthening of RC structures (fib 2001; ISIS 2001; ACI-440.2R 2002, 2008; CNR-DT200 2004; Concrete Society 2004); that is, all these design provisions are limited to designs of FRP jackets for short columns, for which the second-order effect is negligible. This limitation can be attributed to the limited amount of research on the behaviour of slender FRP-confined RC columns [17].

An experimental study was recently completed by the authors on the mechanical properties of CFGT columns with various slenderness ratios and eccentricities. The detailed experimental program and the test results are presented and discussed in the following sections. The main objective of this study was to assess the general behaviour of CFGT columns with slenderness ratios ranging from 14 to 48 and eccentric ratios ranging from 0.1 to 1.0. Additionally, based on the experimental data and the data in the literature, the reduction coefficients of slenderness ratio and eccentricity were determined via parameter identification method. Furthermore, the bearing capacity formula of the CFGT slender columns was established under eccentric loading. A comparison between the test data and the theoretical results showed that the proposed formula was successful.

## 2. Experimental Work

### 2.1. Test Specimens

In total, twelve CFGT circular columns were fabricated. All of the specimens were laterally confined within GFRP tubes. The specimens were divided into two series, series A and series B; six specimens in each series. For series A, the effect of the slenderness ratio was the main focus, and the specimens in were tested under axial loads. For series B, the effect of the eccentricity was the main focus, and the specimens were tested under eccentric loads. All of the specimens had the same height (h = 1000 mm) in series B, whereas in series A, the height of the specimens changed from 600 to 2400 mm. For both series A and series B, specimen 0 was a contrast specimen. The test matrix is presented in Table 1.

### 2.2. Material Properties

Three materials were used for fabricating the test specimens. These materials were concrete, the FRP tubes, and steel reinforcements (deformed steel bars and hoop steels). The following sections provide descriptions of the mechanical properties of the different materials used in this research.

#### 2.2.1. Concrete

All of the specimens were constructed using ready-mixed high strength concrete (HSC). According to the Chinese Standard GB/T50081-2019, the actual compressive strength was determined from the testing of six concrete cylinders “150 mm × 300 mm” on the same day of the testing of the columns. The average compressive strength and tensile strength of the concrete were 64.30 and 6.32 MPa, respectively.

#### 2.2.2. Steel Bars

According to the Chinese Standard GB50010–2019, six deformed steel bars (nominal diameter of 12 mm) were used as longitudinal reinforcements, and steel bars with diameters of 6.5 mm were used as hoop reinforcements with 180 mm spacing in all of the CFGT columns. Table 2 presents the mechanical properties of the steel bars.

#### 2.2.3. GFRP Tubes

The GFRP tubes were fabricated using a filament-winding technique and E-glass fiber and epoxy resin, with different fibre angles of ±57.50° with respect to the longitudinal axis of each tube. The internal diameters were equal to 200 mm, and the thickness of every tube was 5 mm. All of the parameters of the GFRP tube were provided by the manufacturer and not tested in the laboratory. Table 3 shows the mechanical properties.

### 2.3. Test Setup and Instrumentation

Figure 1a shows the axial compression schematic and the test specimen inside the testing machine. In order to ensure the hinged connection at each end of the column, two flat hinges were placed on the top and bottom ends of the composite slender column. Several responses were targeted and monitored during the testing. First, the applied axial load and the machine head axial displacement were measured with the machine’s sensitive internal load cells and linear variable differential transformers, respectively [23]. Second, eight strain gauges were mounted on the GFRP tube at mid-height in both the axial and lateral directions to measure the axial and lateral strains. Additionally, five in-plane linear variable displacement transducers (LVDTs) were placed at five levels on each specimen to record the lateral displacements of each column: the top, top quarter, mid-height, bottom quarter, and bottom. 

Figure 1b shows the eccentric compression schematic and the test specimen inside the testing machine. The top and bottom ends of the column specimens to be tested under eccentric axial loads were each capped with a circular loading head that had a 200 mm inner diameter and 100 mm height, made from high strength steel. The loading head consisted of a loading plate that was 50 mm thick and a rigid column cap. The fixed hinge support was located on the loading head. The distance between the fixed hinge support and the centre of the column section was the nominal eccentricity. Through the action of knife edge, the bending direction of a composite column can be ensured [24]. Four strain gauges were mounted on the GFRP tube at mid-height, in both the axial and lateral directions, to measure the axial and lateral strains. Additionally, five in-plane linear variable displacement transducers (LVDTs) were located at five levels on each specimen to record the lateral displacements of each column: the top, top quarter, mid-height, bottom quarter, and bottom.

All of the specimens were loaded using a 5000 kN capacity-testing machine. Top and bottom adjustable steel roller bearings bolted to the top rigid plates of the steel end caps were used to attain the predesignated eccentricity and to replicate the case of a perfect pin-ended column: k = 1. For part of the linear stress–strain testing (up to 75% of the specimen’s estimated capacity), the test was conducted with a load controlled technique at a rate of 2.5 kN/s. Then, the testing continued under displacement control at a displacement rate of 0.002 mm/s until failure [25]. During the test, the load, axial displacements, lateral displacements, and strains were recorded automatically using a data acquisition system connected to a computer.

## 3. Test Results and Discussion

### 3.1. Effects of the Slenderness Ratio on CFGT Column Properties

#### 3.1.1. Failure Modes

Figure 2 shows the overall failure modes of the composite slender columns. The slenderness ratio was the main parameter affecting the modes of failure in this study. FRP tube rupturing and/or column instability were the dominant failure modes for the CFGT specimens, depending on the slenderness ratio. As shown in Figure 2, the failure modes for specimens L0 and L1 were marked by rupturing of the FRP outer tube on the flat side at around mid-height of the specimens due to local stress concentration. Those results are similar to the test observations presented in [26]. It is of interest to mention that for specimens with slenderness ratios less than or equal to 16, the CFGT columns buckled just before the rupturing of the tube. Increasing the slenderness ratio gradually changed the modes of failure to column instability. It is shown in Figure 2 that the ruptures in the FRP tubes all occurred on the flat side. The instability of each specimen was evident in the degree of curvature under the final failure load. Although each column started to buckle at 85% of its failure load, the deflected column was still stable and carried more axial loads. This indicates that these specimens behaved as slender columns. Loading the specimens continued until the specimens could not maintain the axial force applied. The recorded failure modes of the CFGT showed that the greater the slenderness ratio, the more significant the curvature of the specimen. In addition, these observations and conclusions closely agree with the experimental investigations conducted by Abdallah M. H. et al. [18].

#### 3.1.2. Peak Axial Loads of CFGT Composite Columns

The key test results for the long CFGT composite columns are shown in Table 1. The curve of the peak load vs. slenderness ratio is shown in Figure 3 (data in Table 1). As the slenderness ratio of a CFGT composite column increased, the peak axial load decreased continuously. As can be seen in Table 1 and Figure 3, specimens L1, L2, L3, L4, and L5 held roughly 7.14%, 16.25%, 26.33%, 30.96%, and 35.74% smaller peak axial loads than specimen L0 did, respectively, and the reductions were greater than those of ordinary concrete columns. The lower peak axial loads of the composite columns were mainly due to the higher slenderness ratios, which changed the failure mode of the composite columns. That is, the material compression failure of specimen L0 was transformed into the instability failure of specimen L5.

#### 3.1.3. Axial Load-Lateral Deformation Behaviour of CFGT Composite Slender Columns

The lateral displacement data of the specimens under axial compression were obtained with the displacement sensors arranged axially on the composite columns. Figure 4 shows the lateral deformations of the composite columns at 0.2 times the ultimate load, 0.4 times the ultimate load, 0.6 times the ultimate load, 0.8 times the ultimate load, the ultimate load, and the axial load reduced to 0.8 times the ultimate load.

Figure 4 shows that the lateral deformation of each composite column was divided into three stages. The first stage was the slow stage. In this stage, the axial load increased from 0 to 0.6 times the ultimate load, and the lateral deformation of the specimen either increased slowly or did not increase. The second stage was the steady growth stage, in which the axial load ranged from 0.6 times to 1.0 times the ultimate load, and the lateral deformation of the specimen increased rapidly. When the ultimate load was reached, the lateral deformation of the specimen with a large slenderness ratio was less extreme than that of the specimen with a small slenderness ratio. This was mainly due to the low ultimate load of the specimen with the large slenderness ratio and the absolute axial load being small, which led to the lesser lateral deformation of the specimen. The third stage was the unstable failure stage. In this stage, the axial load decreased and the lateral deformation explosively increased. This was mainly due to the influence of the slenderness ratio of the specimen, which led to the change in failure mode. The instability failure occurred in the specimen with a large slenderness ratio. Compared with specimen L0, the instability of specimen L5 was more obvious at the later stage of loading, and the lateral displacement of the specimen increased rapidly after reaching the ultimate load.

Figure 5 depicts the lateral deformation at the middle of every composite slender column under the axial loads. At the ultimate load, the curve changes course from increasing to decreasing. At the same time, the lateral deformation of the specimen with a large slenderness ratio was smaller than that of the specimen with a small slenderness ratio: 13.6 mm for specimen L0 and 5.67 mm for specimen L5. However, after failure, the lateral deformation of the specimen with a large slenderness ratio increased rapidly. When the load decreased to 0.8 times the ultimate load, the lateral displacement of specimen L0 was 19.67 mm, and that of specimen L5 was 38.75 mm. This was mainly due to the fact that the slenderness ratio of specimen L5 was four times that of reference specimen L0. The increase in the slenderness ratio not only reduced the ultimate load of the composite column by 35.74%, but also caused the instability failure of the composite column, which led to a large increase of the corresponding lateral displacement of the composite column.

#### 3.1.4. GFRP Tube Strain under Axial Load

The longitudinal and circumferential strains on each GFRP tube at five levels of height were recorded by strain gauges attached to the GFRP’s surface, as shown in Figure 6. The curves on the left side of Figure 6 represent the circumferential strains of the odd numbered measuring points; and the longitudinal strains of the even numbered measuring points are on the right side of each graph. Both the longitudinal strains and the circumferential strains of the GFRP tubes were much larger than those of the ordinary concrete column. This is mainly because GFRP tubes have a lower elastic modulus and higher ultimate strain. However, the strains corresponding to the ultimate load, whether circumferential or axial, decreased as the slenderness ratio increased. This was mainly due to the reduction of ultimate bearing capacity resulting in the reduction of actual strain. The strains on the left and right sides of the same section varied from the beginning of loading. As the load increased, the difference became more and more prominent. This occurred due to the effect of initial loading eccentricity. As the ultimate bearing capacity was approached, the compressive strain and the ring tension strain of the GFRP tube on the side of greater compressive stress increased rapidly, and the compressive strain and the ring tension strain on the other side increased slightly. Additionally, the strain of the composite column with the slenderness ratio greater than 32 showed negative growth. The average circumferential strain of the GFRP tube was larger than 10,000 με when the slenderness ratio was less than 32. For the composite column with a large slenderness ratio, the longitudinal buckling of the column was large, the circumferential strain of the GFRP tube was small, and the constraint effect of the tube on the concrete was very small, which was close to the situation of the unconstrained reinforced concrete column. The above strain changes were due to the instability failure of the composite columns, which led to bending. This made the stress inconsistent—and accordingly, the strain changes-on the two sides of the same section.

### 3.2. Effects of the Eccentricity on CFGT Column Properties

#### 3.2.1. Failure Modes

Figure 7 shows the overall failure modes of the composite columns under eccentric loads. The specimens with lower eccentricity failed abruptly by the rupturing of their FRP tubes in the hoop direction. The GFRP tube had a larger fracture zone. With higher eccentricity, separations were observed on the FRP tube among the horizontal FRP layers on the tension side, which is in line with the test observations presented in [27]. This happened because the second-order effect of bending moment caused by eccentricity changed the failure mode of composite columns from compression failure to tensile failure.

#### 3.2.2. Ultimate Bearing Capacity of the CFGT Columns under Eccentric Loads

Figure 8 shows the variations in the bearing capacity with the eccentricity. It can be seen from Figure 8 that the bearing capacity decreased as the eccentricity increased, and the speed of the reduction was first fast, and then slow. Compared with specimen BC0-H, specimens BC1-H, BC2-H, BC3-H, BC4-H, and BC5-H had 49.99%, 77.42%, 89.06%, 93.45%, and 94.44% lower peak loads (CFGT columns). Because the lateral constraint of the concrete decreased as the eccentricity increased, the influence of the concrete’s strength on the bearing capacity decreased gradually, the influence of the longitudinal reinforcement on the composite column obviously increased for the peak load, and the influence of the concrete’s strength on the bearing capacity decreased gradually. Additionally, the GFRP tubes could only affect the strength of the core concrete. Therefore, the larger the eccentricity, the worse the constraint effect, and the lower the peak load.

#### 3.2.3. Lateral Deformation Behaviour of CFGT Composite Columns under Eccentric Loads

The relationship between the axial load and lateral deformation is shown in Figure 9. As can be seen from Figure 9, the lateral displacements of the specimens did not exceed 3 mm at 0.2 times the respective ultimate loads. As the axial load increased, the lateral displacement increased. Before the axial load reached 0.4 times the ultimate load, the lateral displacement increased somewhat, but once the axial load exceeded 0.4 times the ultimate load, the lateral displacement increased rapidly. The greater the eccentricity, the greater the lateral displacement. The lateral displacement of specimen BC2-H increased by 94.70% compared to that of specimen BC0-H at the ultimate load. The lateral displacement of specimen BC5-H increased by 209.70% compared to that of specimen BC0-H. These differences were mainly due to the effect of the GFRP tube decreasing gradually, and the effect of the longitudinal reinforcement increasing gradually. As the eccentricity increased, the failure mode of the composite column changed from compression failure to bending failure.

#### 3.2.4. GFRP Tube Strain under an Eccentric Load

Figure 10 depicts the relationships between the axial load and the GFRP tube strain of the CFGT columns. It can be seen in Figure 10 that the load–strain curve of the CFGT column was close to the centre-fold line. The longitudinal ultimate compressive strain of the GFRP tube was much larger than that of ordinary concrete. The maximum longitudinal ultimate compressive strain of the GFRP tube reached 20,000 με, which is much larger than 3300 με for ordinary concrete. As the eccentricity increased, both the longitudinal and circumferential ultimate strains decreased, and the amplitude of the decrease reached more than 50%. The bending moment of the GFRP tube increased significantly due to the increase of the eccentricity, and the bending resistance of the GFRP tube was poor, so the failure occurred earlier.

## 4. Calculation Method of Ultimate Bearing Capacity

Based on the experimental results and phenomena, the failure form of the CFGT column gradually changed from material failure to overall instability with the increase of the slenderness ratio and the eccentricity. As a result, the ultimate bearing capacity decreased rapidly, which was very different from the calculation formula of the bearing capacity for the CFGT short column. Therefore, the influences of the slenderness ratio and the eccentricity had to be considered in the calculation of the CFGT slenderness columns under an eccentric load.

### 4.1. Slenderness Ratio Reduction Factor of the CFGT Column

Due to the existence of defects such as the initial eccentricity and initial bending, the axial compression columns with larger slenderness ratios produced lateral deflection at the initial stage of loading, and then produced a second-order effect and an additional bending moment. Finally, the concrete column was destroyed by bending instability, and the ultimate bearing capacity was lower than that of the short column. The same problem still existed for the CFGT columns, but the mechanical properties of the CFGT columns were more complicated. The stability bearing capacity of the concrete long columns was analysed with the following two methods: one method was the Euler formula calculation model based on tangent modulus theory, and the other method was the empirical formula based on the regression of test results. In this research, the reduction coefficient formula of the slenderness ratio was obtained from parameter identification method of the experimental data and some literature [19,28,29]:(1)$$\varphi_{l}=1-\delta \sqrt{\frac{l}{d}-4}$$

In the formula:

$\varphi_{l}$—reduction coefficient of slenderness ratio; for l/d<4.0,${\;\varphi }_{l}\;$= 1.0;

l—The calculated length of the columns

d—the outer diameter of concrete column

δ represents the coefficient. When the slenderness ratio was less than 20, δ = 0.15. When the slenderness ratio was greater than 36, δ = 0.18. The intermediate values were obtained with the interpolation method.

### 4.2. Eccentricity Reduction Coefficient of the CFGT Column

The effect of the eccentricity on the ultimate bearing capacity of the CFGT columns was more significant, and this effect was obtained from the experiments described in this paper and some literature references. Using the empirical formula of the concrete-filled steel tubular columns and the test data in this research and some literature [13,29,30,31], the formula of the eccentric compressive bearing capacity of the CFGT composite column under an eccentric load was obtained with regression. The formula of the eccentricity reduction coefficient is as follows:(2)$$\varphi_{e}=\frac{1}{1+c\frac{e_{0}}{r_{c}}}$$
where $e_{0}$ is the eccentricity, $r_{c}$ is the outer radius of the section of the member, and c is the coefficient. For $e_{0}/r_{c}\leq 0.4$, c = 2.96; for $e_{0}/r_{c}\geq 0.8$, c = 6.18. The intermediate value could be calculated with the linear difference.

The parameter c value in the eccentricity reduction coefficient was affected not only by the eccentricity but also by the thickness of the GFRP tube, the mechanical properties of the fibre and the base, the fibre winding angle, and so on. Therefore, parameter c had to be a function of the variation of the thickness, winding angle, and fibre content. Due to the limitation of the experimental research described in this paper, a more in-depth study of the factors affecting the value of parameter c was not performed, but the research methods and the experimental data in this research could provide a useful reference for future researchers.

### 4.3. Simplified Calculation Method of Bearing Capacity for CFGT Slender Columns with Eccentric Load

The bearing capacity of CFGT slender columns under an eccentric load is mainly affected by two factors: slenderness ratio and eccentricity. Therefore, when calculating the ultimate bearing capacity, the influences of the slenderness ratio and the eccentricity on the ultimate bearing capacity were considered. The formula is as follows: (3)$$N_{u}=\varphi_{l}\varphi_{e}N_{0}$$
where Nu is the ultimate bearing capacity of composite slender columns under an eccentric load, $\varphi_{e}$ is the eccentric reduction factor,$\;\varphi_{l}$ is the slenderness ratio reduction factor, and N_0_ is the ultimate bearing capacity of composite short columns.
(4)$$N_{0}{=f}_{\mathrm{cc}}^{\prime }A_{\mathrm{cor}}$$
(5)$$f_{\mathrm{cc}}^{\prime }=f_{\mathrm{cs}}^{\prime }+{6.5f}_{\mathrm{con}}{}^{0.72}=f_{\mathrm{cs}}^{\prime }\left(1+\frac{6.5}{f_{\mathrm{con}}{}^{0.28}}\left(\frac{f_{\mathrm{con}}}{f_{\mathrm{cs}}^{\prime }}\right)\right)$$
(6)$$f_{\mathrm{con}\left(\mathrm{FRP}\right)}=\frac{f_{\mathrm{su}}t}{d_{0}/2}$$
(7)$$f_{\mathrm{cs}}^{\prime }=\frac{f_{\mathrm{yl}}A_{s}{+f}_{\mathrm{co}}^{\prime }}{\left(A_{s}{+A}_{c}\right)}=\left(f_{\mathrm{yl}}{-f}_{\mathrm{co}}^{\prime }\right)\rho_{s}{+f}_{\mathrm{co}}^{\prime }$$

$f_{\mathrm{cc}}^{\prime }$ = axial compressive strength of confined concrete.

$A_{\mathrm{cor}}=$ area of core concrete in GFRP tube.

$f_{\mathrm{con}}$ = restraint stress for GFRP tube and hoop steel on core concrete.

$f_{\mathrm{cs}}$ = conversion strength to concrete section.

$f_{\mathrm{su}}$ = circumferential tensile strength for GFRP tube.

t = wall thickness.

$d_{0}$ = inner diameter of GFRP tube.

$f_{\mathrm{yl}}$ = the yield strength of the longitudinal reinforcement.

$f_{\mathrm{co}}$ = axial compressive strength of unconstrained concrete cylinder.

### 4.4. Verification of the Calculation Formula

In order to verify the correctness of the formula proposed in this paper, data in the literature were introduced, and calculations were performed with the formula suggested in this paper and then compared with the experimental values. The results of the comparison are shown in Table 4. The error interval distribution map was drawn according to Table 4, as shown in Figure 11. In Table 4 and Figure 11, it can be seen that the calculated results are in good agreement with the experimental results. The errors of 13 specimens were less than 5%, accounting for 39% of the total number of specimens. The errors of 23 specimens were not greater than 10%, accounting for 69% of the total. Only one specimen’s error was over 20%, and that error was 36.5%. Therefore, it can be considered that the test error was not greater than 20%, which proves the validity and applicability of the formula.

## 5. Conclusions

This paper reports the results of an experimental investigation into the influences of specimen slenderness and eccentricity on the compressive behaviour of CFGT columns. Based on the results and discussion presented in this paper, the following conclusions can be drawn:

Specimens with low slenderness ratios often failed with continuous rupturing of the FRP shell from the top to the bottom, whereas specimens with high slenderness ratios displayed only localized segmented rupturing, independent of the concrete strength. As the slenderness ratio increased from 12 to 48, the strength dropped rapidly from about 93% of the equivalent short column to <65%.

The behaviour of the CFGT column was significantly affected by the eccentricity. Different eccentricity could cause the CFGT column to fail in the tension or compression modes. The CFGT columns tested under concentric or low (e/D = 10%) or moderate (e/D = 30%) eccentric loads exhibited compression failure—specifically concrete cover spalling at peak load, followed by a significant drop in column carrying capacity. Additionally, separations were observed on the FRP tube, among the horizontal FRP layers, on the tension side, for the specimens under high eccentric loads. Compared with specimen BC0-H, specimens BC1-H, BC2-H, BC3-H, BC4-H, and BC5-H had 49.99%, 77.42%, 89.06%, 93.45%, and 94.44% lower peak loads for the CFGT columns. This was mainly because the GFRP tubes could only affect the strength of the core concrete. Therefore, the larger the eccentricity, the worse the constraint effect, and the lower the peak load.

The slenderness ratio and the eccentricity had significant influences on the ultimate axial strain and the hoop strain, with increases in the H/D ratio and the eccentricity causing decreases in the strain enhancement ratio.

Based on the experimental results and the database from the literature, a modified equation was proposed to predict the bearing capacity of this type of CFGT column, and the predicted values were found to be in good agreement with the experimental results. The errors of 13 specimens were less than 5%, accounting for 39% of the total number of specimens. The errors of 23 specimens were not greater than 10%, accounting for 69% of the total. Only one specimen’s error was over 20%, and that error was 36.5%. Therefore, it could be considered that the test error was not greater than 20%, which proved the validity and applicability of the formula.

The conclusions in this paper were based on experimental investigations of twelve specimens. Thus, more experiments need to be conducted to fully validate the behaviour of the CFGT columns under different loading conditions.

## Figures and Tables

**Figure 1 polymers-13-02968-f001:**
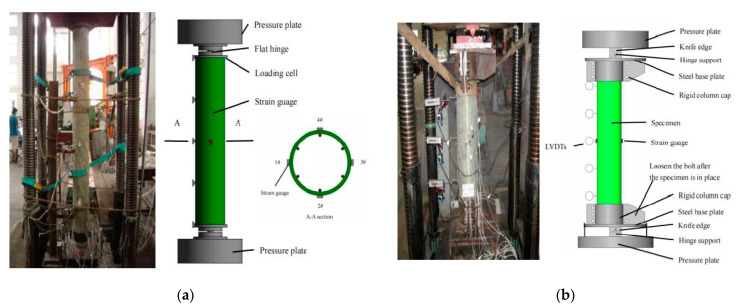
Test setup and instrumentation: (**a**) A composite slender column under an axial load; (**b**) A CFGT composite column under an eccentric load.

**Figure 2 polymers-13-02968-f002:**
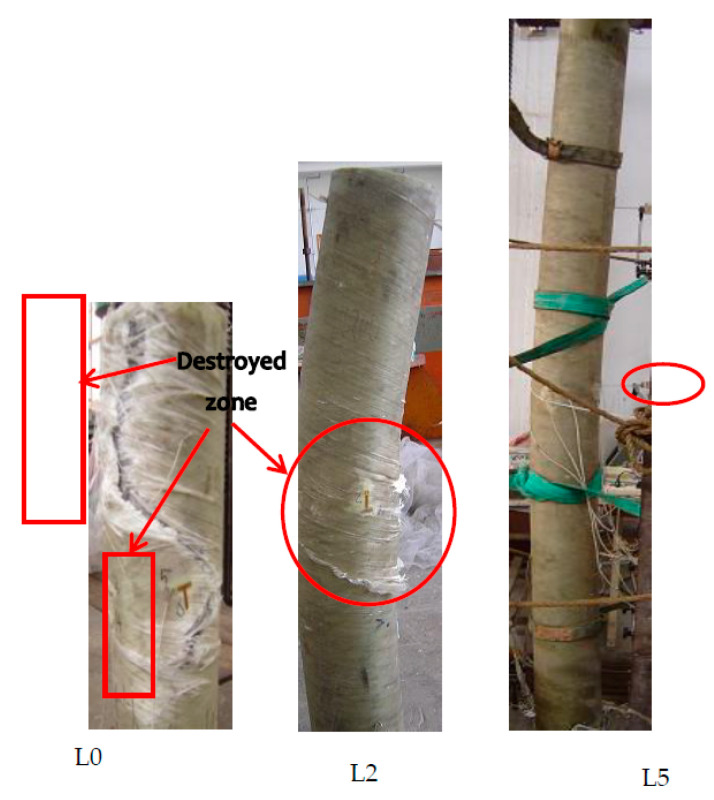
Typical failure modes of composite slender columns.

**Figure 3 polymers-13-02968-f003:**
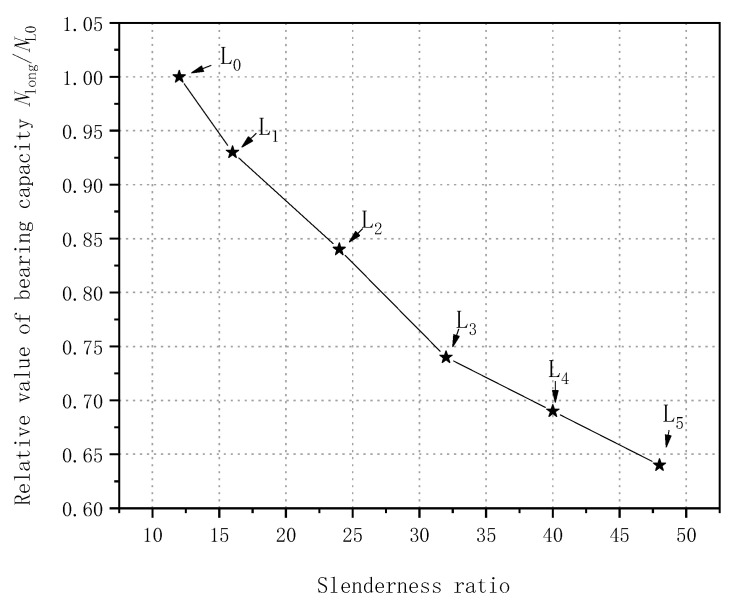
Peak load–slenderness ratio relationship for CFGT composite columns.

**Figure 4 polymers-13-02968-f004:**
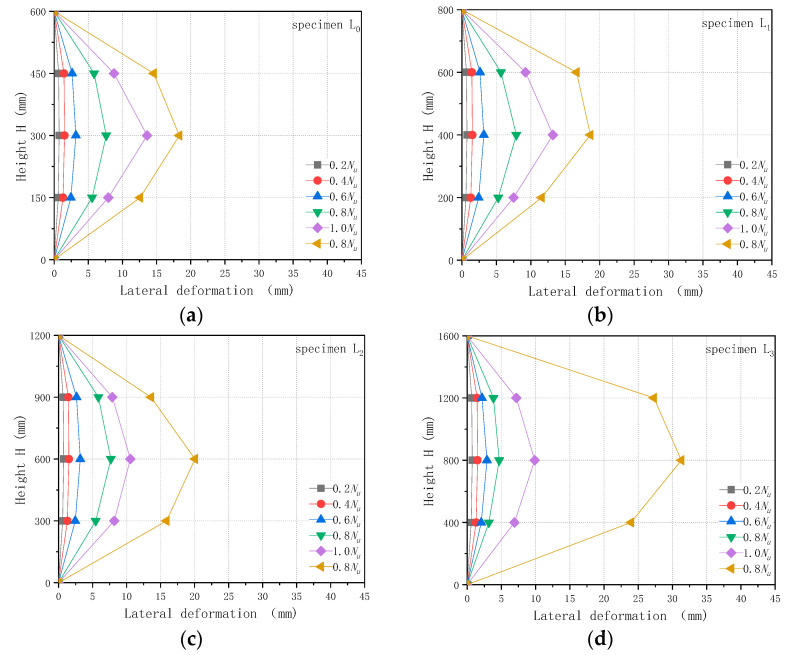
Peak load-slender ratio relationship for CFGT composite columns: (**a**) L0; (**b**)L1; (**c**) L2; (**d**) L3; (**e**) L4; (**f**) L5.

**Figure 5 polymers-13-02968-f005:**
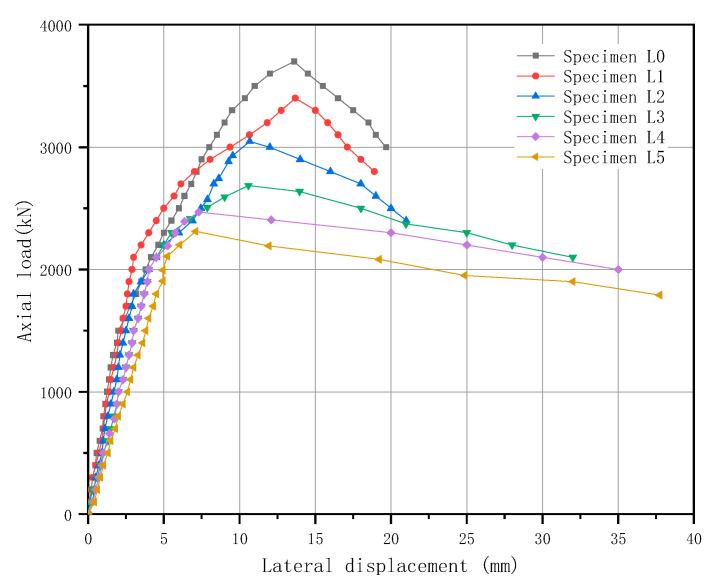
Axial load–lateral deformation curves of composite slender columns, measured at mid-height.

**Figure 6 polymers-13-02968-f006:**
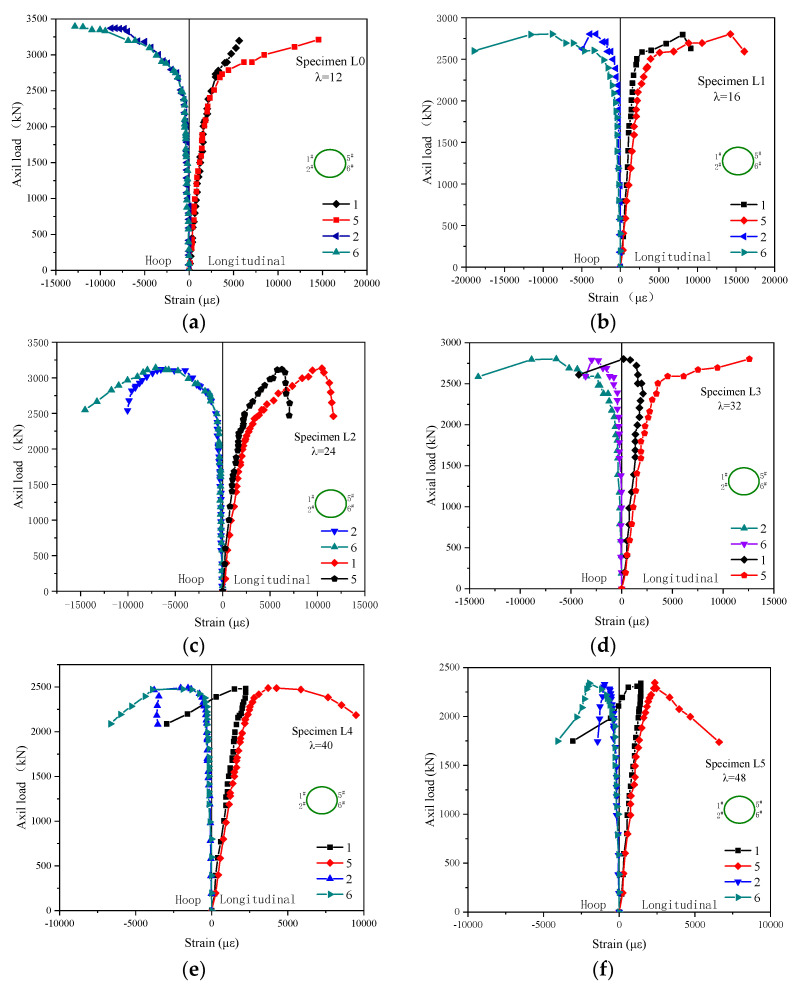
The relationship between the GFRP tube strain and axial load: (**a**) L0; (**b**)L1; (**c**) L2; (**d**) L3; (**e**) L4; (**f**) L5.

**Figure 7 polymers-13-02968-f007:**
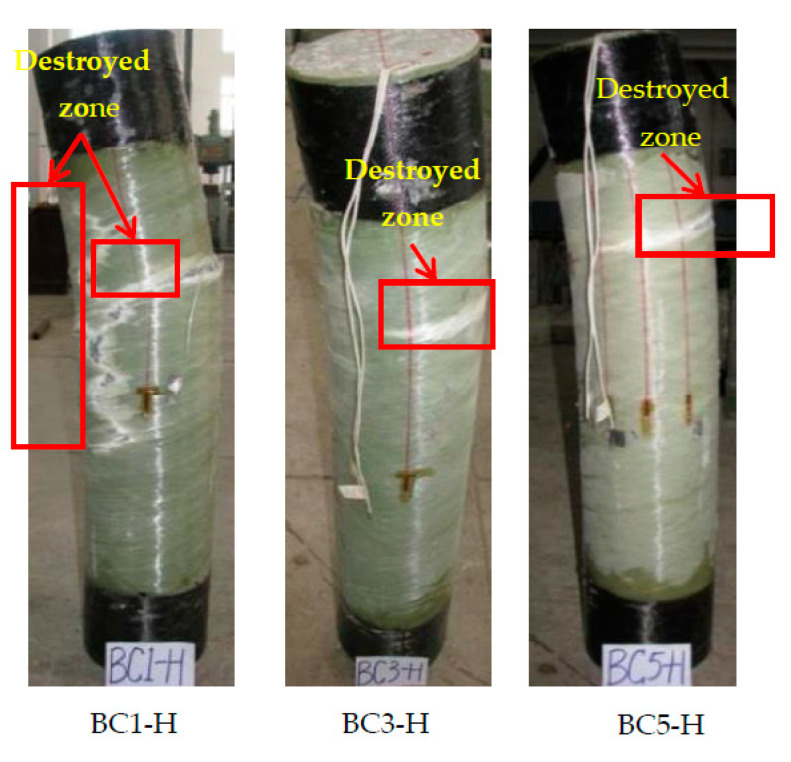
Typical failure modes for the eccentrically loaded specimens.

**Figure 8 polymers-13-02968-f008:**
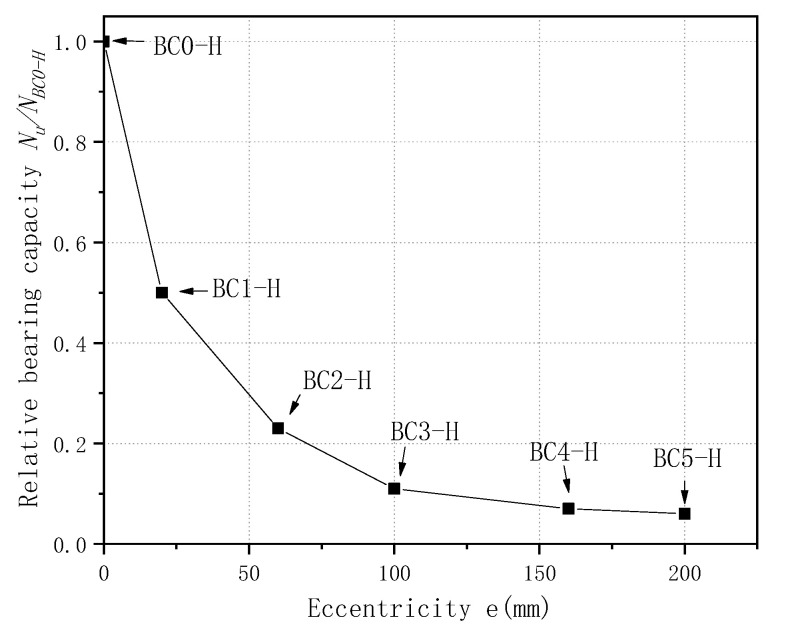
The curve relative bearing capacity vs. eccentricity.

**Figure 9 polymers-13-02968-f009:**
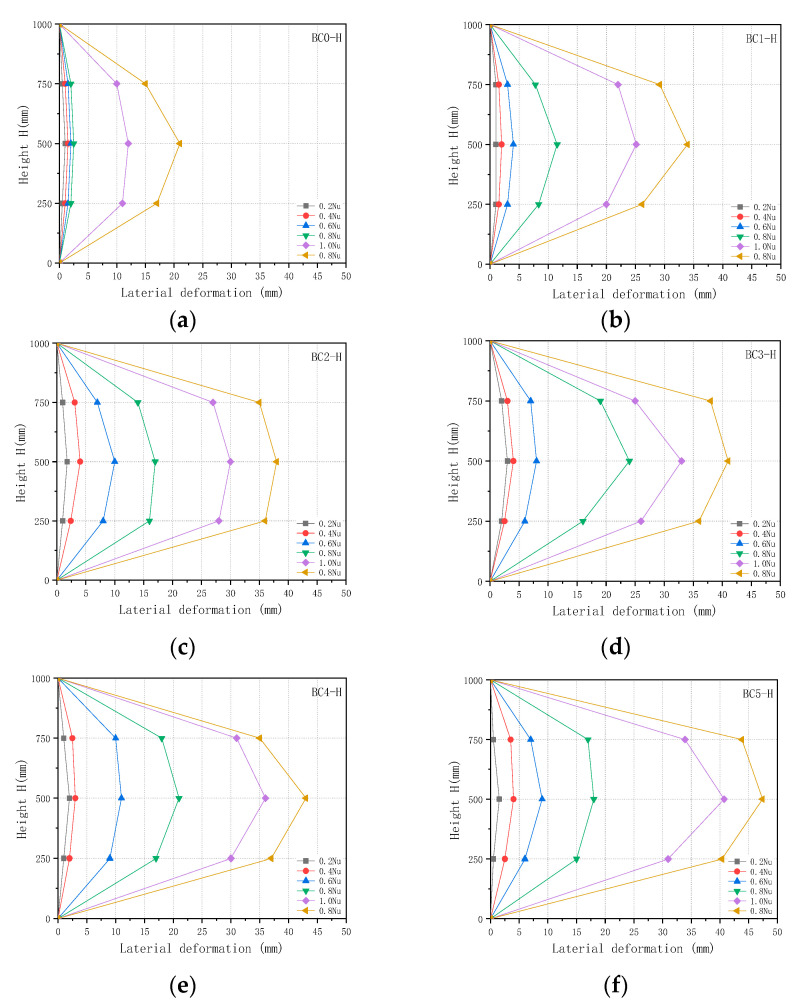
Lateral deformation vs. height with different loads: (**a**) BC0-H; (**b**)BC1-H; (**c**) BC2-H; (**d**) BC3-H; (**e**) BC4-H; (**f**) BC5-H.

**Figure 10 polymers-13-02968-f010:**
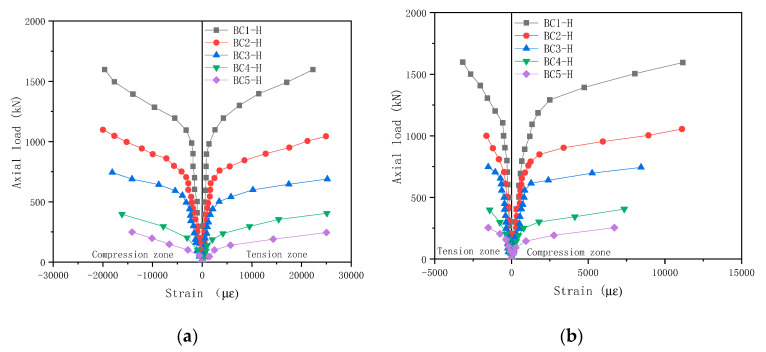
Axial load–strain relationship curves for CFGT columns with eccentric loads (**a**,**b**).

**Figure 11 polymers-13-02968-f011:**
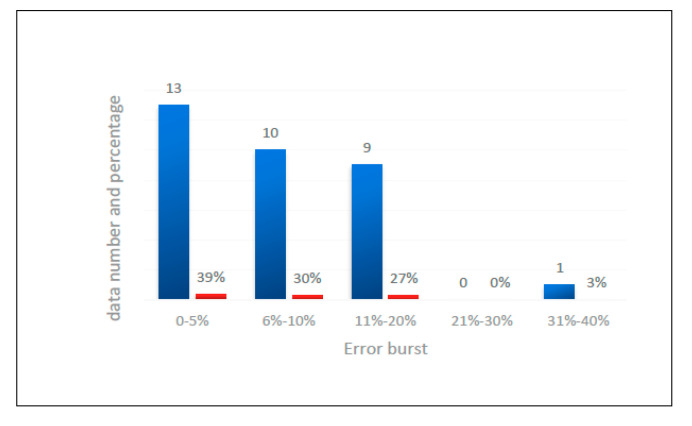
Calculation error distribution.

**Table 1 polymers-13-02968-t001:** The test matrix and the test results of column specimens.

Serial	Specimen ID	Height l_0_ (mm)	Slenderness Ratio λ	Eccentricity (mm)	Eccentricity Ratios e/D	Peak Load (kN)	Reduction in Peak Load (%)	Lateral Deformation at Peak Load (mm)
Series A	L0	600	12	0	0	3720.27	-	13.60
	L1	800	16	0	0	3454.50	7.14	13.21
	L2	1200	24	0	0	3115.70	16.25	10.56
	L3	1600	32	0	0	2740.67	26.33	9.86
	L4	2000	40	0	0	2568.64	30.96	6.43
	L5	2400	48	0	0	2390.63	35.74	5.67
Series B	BC0-H	1000	0	0	0	3288.80	-	13.03
	BC1-H	1000	20	20	0.1	1644.75	49.99	25.37
	BC2-H	1000	20	60	0.3	742.54	77.42	30.02
	BC3-H	1000	20	100	0.5	359.69	89.06	33.93
	BC4-H	1000	20	160	0.8	215.36	93.45	36.35
	BC5-H	1000	20	200	1.0	183.46	94.44	40.35

Note: λ = 4*l*_0_/D, where $l_{0}$ indicates the calculated length of the slender column for the case of a hinge at both ends in this test. D represents the GFRP tube inner diameter, e is the eccentric distance, and e/D is the eccentricity ratio.

**Table 2 polymers-13-02968-t002:** Mechanical properties of the steel bars.

Reinforcement Type	Nominal Diameter (mm)	Tensile Modulus of Elasticity (GPa)	Yield Strength (MPa)	Ultimate Strength (MPa)	Yield Strain (%)
Hoop reinforcement	6.5	210	355	415	0.20
longitudinal reinforcement	12	200	385	509	0.18

**Table 3 polymers-13-02968-t003:** The mechanical properties of the GFRP tubes.

Circumferential Tensile Strength (MPa)	Axial Tensile Strength (MPa)	Hoop Elastic Modulus (MPa)	Axial Elastic Modulus (MPa)
430	156	24,610	9760

**Table 4 polymers-13-02968-t004:** The results of the comparison.

Source of Data	Specimen Number	Length (mm)	Diameter (mm)	Height-to-Diameter L/D	Slenderness Ratio kL/D	Eccentricity (mm)	Slenderness Reduction Factor	Eccentric Reduction Factor	$N_{exp}^{*}$	$N_{cal}^{**}$	$\frac{N_{exp}^{*}}{N_{cal}^{**}}$
Literature [28]	1	305	147.3	2.1	4	0	1	1	97.41 MPa	96.36 MPa	1.011
	2	813	147.3	5.5	11	0	0.82	1	79.98 MPa	78.66 MPa	1.017
	3	1372	147.3	9.3	18	0	0.65	1	60.27 MPa	63.08 MPa	0.955
	4	1651	147.3	11.2	22	0	0.52	1	49.18 MPa	49.82 MPa	0.987
	5	2286	147.3	15.5	30	0	0.39	1	38.08 MPa	37.54 MPa	1.014
	6	2591	147.3	17.6	34	0	0.34	1	34.76 MPa	32.40 MPa	1.073
	7	2743	147.3	18.6	36	0	0.31	1	27.90 MPa	30.09 MPa	0.927
Literature [18]	8-S-I	610	152	4	8	0	1.00	1	1652 kN	1678 kN	0.985
	12-S-I	912	152	6	12	0	0.79	1	1454 kN	1322 kN	1.100
	16-S-I	1216	152	8	16	0	0.70	1	1202 kN	1175 kN	1.023
	20-S-I	1500	152	10	20	0	0.63	1	1127 kN	1061 kN	1.062
Literature [13]	GT-0	812	203	4	16	0	1	1	1884 kN	1797 kN	1.048
	GT-25	812	203	4	16	25	1	0.503	860 kN	903 kN	0.952
	GT-50	812	203	4	16	50	1	0.307	523 kN	551 kN	0.949
Literature [30]	FTRC-0	800	240	3.33	6.67	0	1	1	1850 kN	2050 kN	0.902
	FTRC-25	800	240	3.33	6.67	25	1	0.619	1474 kN	1268 kN	1.162
	FTRC-50	800	240	3.33	6.67	50	1	0.371	1038 kN	760 kN	1.365
Literature [31]	GRC-2	700	200	3.5	7	20	1	0.628	1344 kN	1117 kN	1.203
	GRC-3	700	200	3.5	7	40	1	0.458	841 kN	814 kN	1.033
	GRC-4	700	200	3.5	7	0	1	1	1548 kN	1778 kN	0.871
Literature [29]	Z1-L-5-40	1000	200	5	20	40	0.85	0.387	1001 kN	993 kN	1.008
	Z2-L-6-40	1200	200	6	24	40	0.75	0.387	942 kN	871 kN	1.081
	Z3-L-7-40	1400	200	7	28	40	0.69	0.387	785 kN	804 kN	0.976
	Z4-L-9-40	1800	200	9	36	40	0.60	0.387	719 kN	698 kN	1.030
	Z6-H-7-0	1400	200	7	28	0	0.69	1.000	2526 kN	2671 kN	0.946
	Z6-H-7-40	1400	200	7	28	40	0.69	0.387	1087 kN	1034 kN	1.051
	Z6-H-7-60	1400	200	7	28	60	0.69	0.247	756 kN	661 kN	1.144
	Z6-H-7-100	1400	200	7	28	100	0.69	0.139	414 kN	372 kN	1.113
	Z6-H-7-160	1400	200	7	28	160	0.69	0.092	237 kN	245 kN	0.966
	Z6-H-9-40	1800	200	9	36	40	0.60	0.387	1005 kN	898 kN	1.120
	Z6-H-9-160	1800	200	9	36	160	0.60	0.092	188 kN	213 kN	0.883
	Z6-H-5-40	1000	200	5	20	40	0.85	0.387	1435 kN	1277 kN	1.124
	Z6-H-5-160	1000	200	5	20	160	0.85	0.092	267 kN	303 kN	0.881
	Z6-H-6-40	1200	200	6	24	40	0.75	0.387	1269 kN	1120 kN	1.133
	Z6-H-6-160	1200	200	6	24	160	0.75	0.092	245 kN	266 kN	0.922

## Data Availability

All the data will be available to the readers.

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
