# Peer review of "Influences of Slenderness and Eccentricity on the Mechanical Properties of Concrete-Filled GFRP Tube Columns"

_polymers, 2021, doi:10.3390/polym13172968_

Round 1
Reviewer 1 Report
This paper reports the results of an experimental investigation into the influence of specimen slenderness and eccentricity on the compressive behavior of CFGT columns. The results indicated that the influence of the slenderness and eccentricity on the composite columns was found to be significant regarding the axial strain, hoop strain, ultimate bearing capacity, lateral displacement, and failure mode.
The paper is sufficiently well written, and the main ideas are sufficiently well developed. In my opinion, the paper should be accepted after strong revisions, based on the following general and specific suggestions.
General suggestions are:
- What are the lessons learned from the conclusions other than the observation?
- Is this finding original to this paper or is it a validation of a previously established notion/fact (see the last observation of the next bullet points)?
Specific suggestions are:
- The use of non-technical words should be avoided.
- Please avoid, as far as possible, the use of abbreviations in the abstract.
- The main aim of selected methods (advantages/disadvantages) should be presented more clearly.
- Please, check the citations.
- The quality of the figures should be improved.
- The well-known formulation should be shortened, and related references should be cited for related researchers.
- More information should be given about the boundary condition.
- Some philosophical discussions may also be included regarding the findings.
- The paper is not prepared according to the journal guidelines.
Reviewer 2 Report
This study aims to measure the effect of Slenderness and Eccentricity on the Mechanical Properties of concrete-filled GFRP Tube Columns. Based on the reviewer's opinion, the following comments should be addressed by authors:
- The introduction section should be extensively improved. The gap of previous investigations that results in doing the current study should be explained in details
- The novelty of this study is not clear and should be explained in details
- The standards used in sections 2.2.1 and 2.2.2 should be referenced.
- Please modify Fig. 2
- Please draw Fig. 3 using curve fitting
- Test setup should be discussed in more detail. Some information such as the importance of Knife edge, the rate of applied load and so on are missed.
- Please use references to clarify your presented test setup
- The authors only reported results with no justification. So, please discuss the behaviours of specimens
- In the results section, the previous papers should be references to confirm the presented results
- Please draw Fig. 8 using curve fitting
- The conclusion section should be modified. Both qualitative and quantitative reviews should be provided
Round 2
Reviewer 1 Report
All the suggestions are correctly addressed, the paper is accepted as is
Reviewer 2 Report
The reviewer appreciate authors for addressing all comments. The paper has been significantly improved. So, the paper is recommended for the publication in the current format.